# Spatial Modeling of Air Pollution Using Data Fusion

Adrian Dudek and Jerzy Baranowski *

Department of Automatic Control & Robotics, AGH University of Science & Technology, 30-059 Kraków, Poland; addudek@agh.edu.pl
* Correspondence: jb@agh.edu.pl

**Abstract:** Air pollution is a widespread issue. One approach to predicting air pollution levels in specific locations is through the development of mathematical models. Spatial models are one such category, and they can be optimized using calculation methods like the INLA (integrated nested Laplace approximation) package. It streamlines the complex computational process by combining the Laplace approximation and numerical integration to approximate the model and provides a computationally efficient alternative to traditional MCMC (Markov chain Monte Carlo) methods for Bayesian inference in complex hierarchical models. Another crucial aspect is obtaining data for this type of problem. Relying only on official or professional monitoring stations can pose challenges, so it is advisable to employ data fusion techniques and integrate data from various sensors, including amateur ones. Moreover, when modeling spatial air pollution, careful consideration should be given to factors such as the range of impact and potential obstacles that may affect a pollutant's dispersion. This study showcases the utilization of INLA spatial modeling and data fusion to address multiple problems, such as pollution in industrial facilities and urban areas. The results show promise for resolving such problems with the proposed algorithms.

**Keywords:** INLA; data fusion; air pollution; spatial modeling

## 1. Introduction

Air pollution is a significant environmental and public health issue, affecting the well-being of millions of people worldwide. The increased industrialization and urbanization of modern society have led to a significant rise in air pollution levels, posing a severe threat to human health and the natural environment [1]. Exposure to air pollution has been linked to a wide range of health effects, including respiratory diseases, cardiovascular diseases, and even cancer.

Given the significant impacts of air pollution on human health and the environment, there is a critical need for data-driven solutions to address this problem. In recent years, there has been a surge of interest in data science approaches to air pollution research, including the use of machine-learning algorithms to model pollution levels and identify sources of pollution with gathered data [2–4]. Choosing the right models and collecting sufficient data are key to solving the problem.

One way to model air pollution data is through statistical models [5]. This allows for solving the problem of the air impact in the space in which it is located and measured. The distance between observations can be a source of correlation for such a model. For instance, housing values in a city typically show a gradual shift between adjacent neighborhoods [6]. Similarly, pollution levels display a spatially smooth pattern, such that measurements taken in close proximity are expected to yield comparable results [7].

These examples all share a common feature: their data are spatially dependent, with observations in close proximity likely to display analogous values and a high degree of spatial autocorrelation. Spatial models account for this autocorrelation to distinguish the general trend, typically conditioned on covariates, from the spatially random variation [8]. Problems that use spatial modeling fall into a variety of categories. One example could

be the problem of land surface temperature (LST). Its trends are measured using space sensors, specifically thermal infrared and passive microwave observations from satellite missions [9]. Satellites are useful in modeling air quality. In [10], we can see satellite-based atmospheric composition data collected by governments for air quality regulation and site-specific legislation compliance. The use of the TROPOspheric Monitoring Instrument showcases the potential of current instruments for detecting and isolating $NO_2$ emissions from regulated point sources. Moreover, satellites are used as a part of the data source for one of the datasets used in our research.

Such spatial modeling can be carried out with use of INLA (integrated nested Laplace approximation) [11,12]. INLA has gained recognition as an efficient approach to Bayesian inference in recent years. It offers a quicker and more user-friendly alternative to other methods, like MCMC, facilitated by the R-INLA package. INLA is designed to handle models expressed as latent Gaussian Markov random fields (GMRFs), but this broad category encompasses numerous practical models. INLA can be easily used for this type of problem, as evidenced by examples in the literature [5,13] in which it has been successfully applied.

Another problem that we often encounter during spatial modeling is the insufficient amount of information obtained from professional observation points. There are only a dozen such stations in the city of our operations. In this case, data fusion comes in handy. The amalgamation of multiple data sources to generate information that is more dependable, precise, and practical than what could be derived from any single source is referred to as data fusion [14]. It allows other data sources, for instance, automotive sensors [15], which include dozens of different sensors [16,17], which are still developing, or multi-sensor satellite images, which can be used to estimate surface soil moisture [18].

Sensors frequently vary in terms of their scale, measurement method, and accuracy. The information gathered through these sensors can be used, for example, to fill in gaps within the data.

In general, data fusion is a field of research that aims to combine information from multiple sources or sensors to produce a more accurate and reliable estimate of a phenomenon or event. With the increasing availability of diverse data sources and the need for improved decision making, data fusion has become an important area of study in many scientific and engineering disciplines [14]. The main challenge in data fusion is to integrate information from heterogeneous sources, which may have different measurement scales, uncertainties, and biases. To address this challenge, several theoretical frameworks and algorithms have been developed that enable the combination of information from multiple sources while accounting for their differences [14].

One such framework is the Bayesian theory of data fusion, which provides a probabilistic approach to combining information from multiple sources. The Bayesian theory assumes that each source of data provides a prior probability distribution over the phenomenon of interest, and the goal is to compute the posterior distribution that combines all the available information. The Bayesian theory also provides a natural way to incorporate expert knowledge and prior beliefs into the fusion process [19].

Another important aspect of data fusion is the selection of appropriate algorithms and techniques for integrating the data. There are several approaches to data fusion, including rule-based methods [20], mathematical modeling, and machine-learning techniques such as neural networks [21] and decision trees [22]. The choice of the fusion method depends on the nature of the data, the characteristics of the sources, and the application requirements.

Data fusion has many applications in fields such as remote sensing, image processing, robotics, health monitoring, environmental monitoring, and weight in motion (WIM). For example, in environmental monitoring, data fusion can combine information from different sensors to produce high-resolution images of the Earth's surface and let us calculate surface soil moisture [18]. In health monitoring, data fusion can be used to integrate data from wearable sensors to monitor the health of patients in real time [23]. In WIM, we can

combine data from different kinds of sensors [24], for instance, piezoelectric sensors [25] with polymer axle load sensors.

Data fusion is a powerful tool for combining information from multiple sources to produce more accurate and reliable estimates. The development of new theoretical frameworks and algorithms for data fusion is expected to play an important role in enabling data-driven decision making in many scientific and engineering domains.

In this paper, we investigate the possibility and effectiveness of using INLA within R-INLA framework in the spatial modeling of air pollution:

- Indoors, with the use of barrier modeling to implement information about obstacles;
- Outdoors, with the use of real data from air quality sensors in the large city;
- With the use of data fusion to enrich input data with datasets from different kinds of sensors and sources.

The key elements of our work are presented in Table 1. It also includes a comparison to other studies in a related field.

**Table 1.** Main contributions of this scientific paper in comparison to other studies in a related field.

| Article | Research Scope | Methodology | Results |
|---------|---------------|-------------|---------|
| Ours | Use of r-INLA package and data fusion for generation of spatial model of air pollution in different scenarios | Instances of modeled indoor and outdoor situations. Indoor case takes advantage of barrier modeling (information about obstacles), and outdoor (city) uses data fusion with different datasets. | Authors successfully created the air pollution model in all cases. Barrier models increase the reliability of indoor model. Data fusion fixes the imperfections of official measurement systems. |
| [5,13] | Use of INLA for spatial prediction of air pollutant concentrations | Creation of multi-pollutant geostatistical models with use of air quality data from monitoring measurements (observed monitoring measurements and sometimes correlations between pollutants). | Prediction accuracy compares favorably to previous efforts to map air pollution. Model properly estimates daily ambient pollution. |
| [6] | Analysis of the Spatial Diversity of Housing Prices | Introduction of composite external effect at a given point in space as the location-value signature (LVS) of that parcel, which is estimated in the model. It plays a role in the determination of the market value of a house residing on that parcel. Model based on fundamental assumptions: interact location variables with attributes like lot size, age, and dwelling size. | Authors use multiple hedonic specifications, including an interactive variables approach. Interactive model significantly improves explanatory power and removes spatial dependence in error terms compared to the noninteractive model within their sample. |
| [21,24,26] | Use of data fusion in different scenarios: vehicle weight, machine learning, classification | In the context of these articles, a similar framework is employed that involves the fusion of data from different sensors or datasets across different scenarios. | By integrating data from diverse sensors or datasets, these frameworks enhance the overall usability and accuracy of models. It allows researchers and practitioners to leverage the strengths of individual data sources and overcome the limitations of each, leading to more robust and reliable results. |

The rest of this paper is organized as follows. First, in the Materials and Methods section, we present the basic theory of INLA, the main points of our algorithm, and the methodology and datasets that we used. Then, we present the results of our experiments, with three different cases of use. The paper ends with a discussion and conclusions.

## 2. Materials and Methods

The main tool used in our modeling considerations is the R-INLA package, which contains detailed implementations of the possibilities of solving problems using sparse matrices, which are described in INLA. INLA, also known as the integrated nested Laplace approximation, was created by Rue, Martino, and Chopin in 2009 to serve as an alternative to traditional Markov chain Monte Carlo (MCMC) methods for approximate Bayesian inference [27]. INLA is specifically designed for models that can be expressed as latent GMRFs due to their computational properties [28]. In the INLA framework, the observed variables $\mathbf{y} = (y_1, \ldots, y_n)$ are modeled using an exponential family distribution, where the mean $\mu_i$ (for observation $y_i$) is linked to the linear predictor $\eta_i$ through an appropriate link function. The linear predictor includes terms on covariates and different types of random effects, and the distribution of the observed variables $y$ depends on some hyperparameters $\theta$. The distribution of the latent effects $x$ is assumed to be a GMRF with a zero mean and precision matrix $\mathbf{Q}(\theta)$, which depends on the hyperparameters $\theta$. The likelihood of the observations, given the vector of latent effects and the hyperparameters, can be expressed as a product of likelihoods for individual observations:

$$\pi(\mathbf{y} \mid \mathbf{x}, \theta) = \prod_{i \in \mathcal{I}} \pi(y_i \mid \eta_i, \theta)$$

In this context, the latent linear predictor is denoted by $\eta_i$ and is one of the components of the vector $x$ that includes all latent effects. The set $\mathcal{I}$ includes the indices of all the observed values of $y$, although some of these values may not have been observed. The primary objective of the INLA approach is to estimate the posterior marginal distributions of the model effects and hyperparameters. This is accomplished by taking advantage of the computational benefits of the GMRF and the Laplace approximation for multidimensional integration [29]. The joint posterior distribution of the effects and hyperparameters can be expressed as:

$$\pi(\mathbf{x}, \theta \mid \mathbf{y}) \propto \pi(\theta)\pi(\mathbf{x} \mid \theta) \prod_{i \in \mathcal{I}} \pi(y_i \mid x_i, \theta)$$
$$\propto \pi(\theta)|\mathbf{Q}(\theta)|^{1/2} \exp\left\{-\tfrac{1}{2}\mathbf{x}^\top \mathbf{Q}(\theta)\mathbf{x} + \sum_{i \in \mathcal{I}} \log(\pi(y_i \mid x_i, \theta))\right\}$$

To simplify the notation, the precision matrix of the latent effects is denoted by $\mathbf{Q}(\theta)$, and its determinant is represented by $|\mathbf{Q}(\theta)|$. Additionally, $x_i = \eta_i$ for values of $i$ that are in the set $\mathcal{I}$ [29]. The computation of the marginal distributions for the latent effects and hyperparameters can be performed by taking into account that:

$$\pi(x_i \mid \mathbf{y}) = \int \pi(x_i \mid \theta, \mathbf{y})\pi(\theta \mid \mathbf{y})d\theta$$
$$\pi(\theta_j \mid \mathbf{y}) = \int \pi(\theta \mid \mathbf{y})d\theta_{-j}$$

It is important to observe that both equations involve integrating over the hyperparameter space and require a reliable approximation of the joint posterior distribution of the hyperparameters. To achieve this, we approximate $\pi(\theta \mid \mathbf{y})$, denoted as $\tilde{\pi}(\theta \mid \mathbf{y})$ [27], and use it to approximate the posterior marginal distribution of the latent parameter $x_i$ as follows:

$$\tilde{\pi}(x_i \mid \mathbf{y}) = \sum_k \tilde{\pi}(x_i \mid \theta_k, \mathbf{y}) \times \tilde{\pi}(\theta_k \mid \mathbf{y}) \times \Delta_k$$

The weights $\Delta_k$ correspond to a vector of hyperparameter values $\theta_k$ in a grid. The calculation of the approximation $\tilde{\pi}(\theta_k \mid \mathbf{y})$ can vary depending on the method used, and it can take on different forms [27].

The main purpose was to create spatial models of air pollution with particulate matter. We divided our research into three separate steps (models):

- A model of an industrial room with a strong source of pollutant emissions (simulated data);
- A barrier model (including obstacles and walls) of an industrial room with a strong source of pollutant emissions (simulated data);
- A city air pollution model (real data).

The first two models, which are based on simulated data, are intended to show the difference between the use and non-use of barrier models that allow the implementation of information about the boundaries in the considered space. In this case, it allows us to model a hall with three rooms, where, in two of them, there are machines that are sources of pollution. The exact plan of the layout of the simulated rooms is shown in Figure 1. In the two larger rooms, there are machines—sources of air pollution. The room at the bottom of the diagram is a vestibule connected by narrow passages with rooms where devices work. This arrangement is to guarantee a lower spread of pollution.

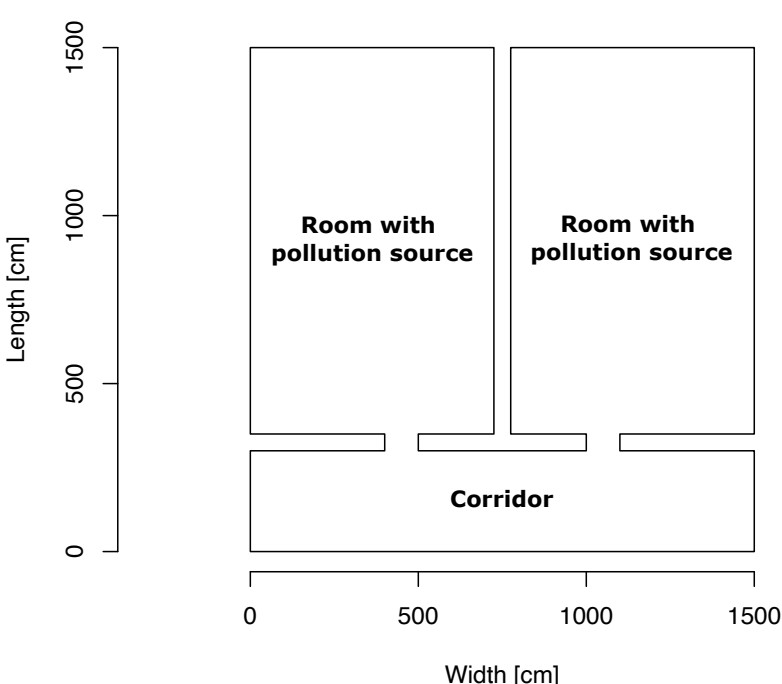

**Figure 1.** Plan of the production hall for the simulation process. The dimensions of the entire room are 15 m width and 15 m length. Inside the two larger rooms, there are machines that are sources of pollution. The room at the bottom of the graph is a vestibule between them, connected by narrow passages. This is to reduce the passage of pollutants between rooms.

We placed two sources of pollution in the defined rooms. Based on these assumptions, we created the distribution of pollutants using a bivariate normal distribution, the results of which can be seen in Figure 2. The obtained distribution simulates the random values of pollutants, where their peak is where the machines are located, i.e., points $(20, 100)$ and $(130, 100)$. Other settings (like the covariation function) have been randomized.

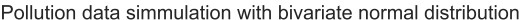

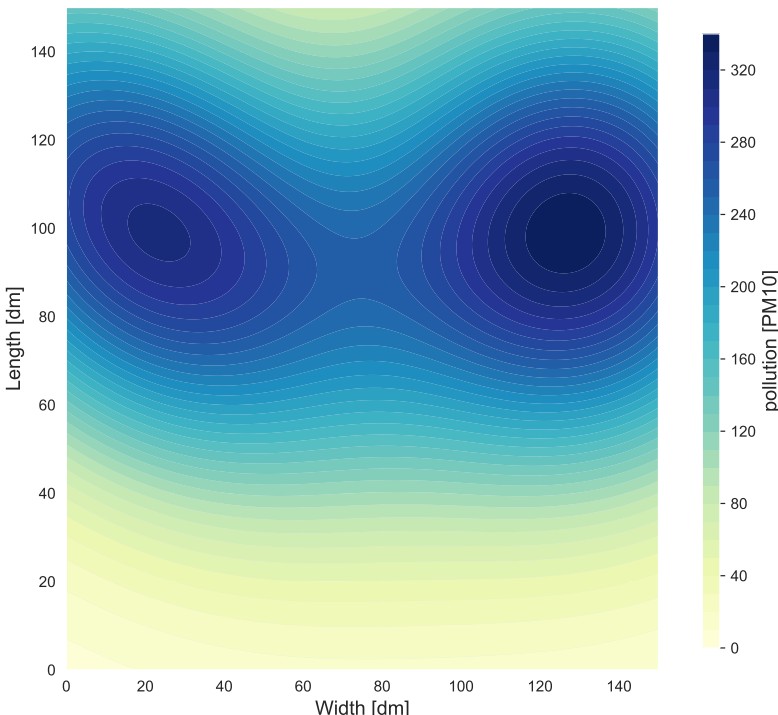

**Figure 2.** Randomly generated data on pollution in the defined production hall of 15 m width and 15 m length. Data were generated with bivariate normal distribution with randomized parameters, with the exception of maxima located at $(20, 100)$ and $(130, 100)$, where pollution sources are located.

The purpose of the third model is to use data fusion from different sources and sensors and create a spatial model of air pollution with INLA in the chosen city. In our case, it is Krakow in Poland. In order to fuse our data, we used two datasets, which provided us with differential sources of data. The first dataset comes from the database of a Polish government organization called the Main Inspectorate of Environmental Protection (MIEP) [30]. Their databases consist of a dozen public stations spread out in the city of Krakow and the surrounding area. Selected stations measured the level of PM10 in the air. These data are available in the MIEP dataset, and detailed information about stations near the city of Krakow can be found in Table 2. In the set of data shown in Table 2, we selected a subset of stations that actually measure PM10 values (bold names).

**Table 2.** All measuring stations near Krakow found in Main Inspectorate of Environmental Protection's dataset. The table shows the basic information to identify each station, such as the international station code and longitude and latitude. In addition, we have provided information on the values measured by the stations. We have highlighted in bold the stations that provide the data needed for the model (PM10).

| Station Name | International Code | Measured Values |
|---|---|---|
| **Krakow, Złoty Róg** | PL0643A | **PM10**, benzo(a)pyrene in PM10 |
| **Krakow, Aleja Krasińskiego** | PL0012A | **PM10**, PM2.5, carbon monoxide, nitric oxide, nitrogen dioxide, nitrogen oxides |
| **Krakow, Dietla** | PL0641A | **PM10**, nitric oxide nitrogen dioxide, nitrogen oxides |
| **Zabierzow, Wapienna** | PL0728A | **PM10**, benzo(a)pyrene in PM10 |

**Table 2.** *Cont.*

| Station Name | International Code | Measured Values |
|:---:|:---:|:---:|
| Krakow, Bujaka | PL0501A | **PM10**, PM2.5, benzo(a)anthracene in PM10, benzo(b)fluoranthene, benzo(k)fluoranthene, cadmium in PM10, dibenzo(a,h)anthracene in PM10, indene(1,2,3-cd)pyrene in PM10, nickel in PM10, nitric oxide, nitrogen dioxide, nitrogen oxides, ozone, benzo(j)fluoranthene in PM10 |
| Krakow, Swoszowice | PL0735A | **PM10**, benzo(a)pyrene in PM10 |
| Skawina, Ogrody | PL0273A | **PM10**, nitric oxide, nitrogen dioxide, nitrogen oxides, sulfur dioxide, benzene |
| Kaszow | PL0640A | Nitric oxide, nitrogen dioxide, nitrogen oxides, ozone |
| Krakow, Piastow | PL0642A | **PM10**, benzo(a)pyrene in PM10 |
| Krakow, Bulwarowa | PL0039A | **PM10**, PM2.5, arsenic in PM10, benzo(a)pyrene in PM10, benzene, cadmium in PM10, carbon monoxide, nickel in PM10, nitric oxide, nitrogen dioxide, nitrogen oxides, lead in PM10, sulphur dioxide |
| Krakow, Wadow | PL0670A | **PM10**, benzo(a)pyrene in PM10, arsenic in PM10, cadmium in PM10, nickel in PM10, lead in PM10 |
| Niepolomice | PL0125A | **PM10**, benzo(a)pyrene in PM10 |
| Szarow | PL0618A | Nitric oxide, nitrogen dioxide, nitrogen oxides, ozone |

The stations used by MIEP are public use stations and intended for the professional use of air quality measurements. The measurement methodology for particulate matter (PM10) is indicated in Directive 2008/50/EC of the European Parliament and of the Council of 21 May 2008 on ambient air quality and cleaner air for Europe (*Official Journal of the European Union*, L 152 of 11 June 2008, p. 1) and in the Regulation of the Minister of the Environment of 13 September 2012 on the assessment of the levels of substances in the air.

The Environmental Protection Inspection tests the content of PM10 in the air using two complementary methods:

- **The gravimetric (reference) method**, which is recognized and used around the world as the most precise measurement method;
- **An automatic method** with demonstrated equivalence to the reference method.

The gravimetric method, also called the manual (reference) method, uses the so-called dust collectors, special devices into which atmospheric air is sucked. Every two weeks, 14 disposable filters are placed in the collector, which the device changes automatically every 24 h. The advantage of this measurement method is its very high accuracy. Its only disadvantage is the time needed to obtain results, which is about 3 weeks. The automatic method has to be equivalent to the reference, so it must be demonstrated that the device meets the requirements for equivalence, and the results of such tests must be submitted to and approved by the European Commission. For measurements performed under MIEP, automatic meters with certificates confirming their equivalence to the reference method are used. These meters continuously measure dust concentrations and are used in the dataset used in our research [31].

In Figure 3, we can see an exemplary station (Dietla) from the set listed in Table 2, located close to the city center of Krakow. In contrast to the described set, the second one coming from the Airly source is a set of various types of more or less professional measuring stations. On the other hand, it has a much larger database that can be fused with the data provided by MIEP.

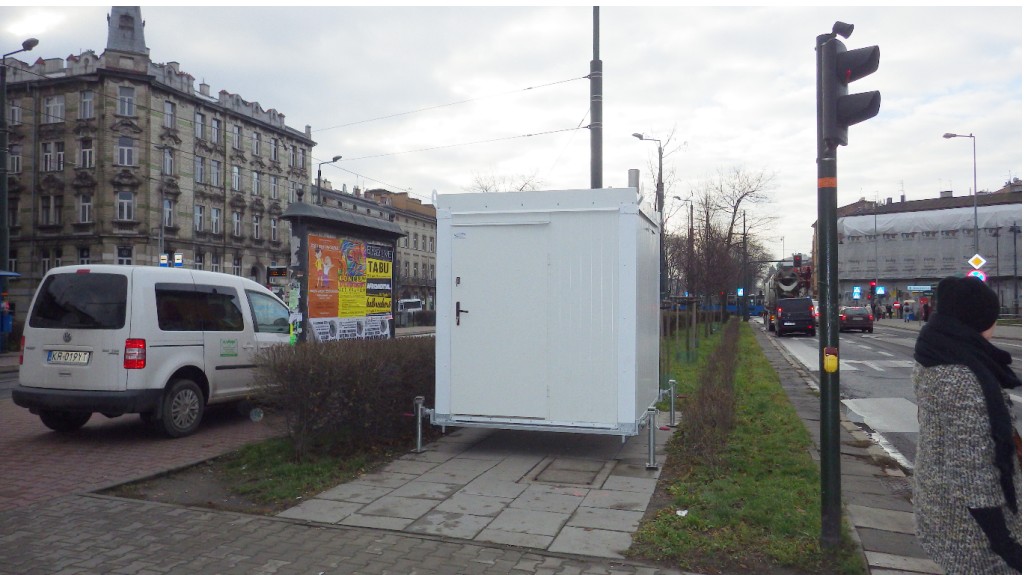

**Figure 3.** Example station used by the Main Inspectorate of Environmental Protection in their measurement system. The station name is Krakow, Dietla (code: PL0641A), and it is located near the city center and can measure the PM10 level, nitric oxide, nitrogen dioxide, and nitrogen oxides. Source: MIEP website [32].

The second set of air quality data, which was used for data fusion, come from the Airly dataset [33]. This dataset consists of a variety of sensors located throughout the city. The information that we collected was for a range of 15 km from the city center of Krakow. The sensors used in this set can be divided into three main types:

1. Air-quality-monitoring reference stations;
2. Low-cost sensors;
3. Satellites.

Reference stations are large enclosures that run on main power and house analyzers capable of generating reference-grade measurements. These structures are fixed at a specific point and provide air quality data. When data from several reference stations are combined, it becomes possible to model air quality for larger urban areas. Reference stations are often employed to ensure compliance with the legal limits of pollutants. There are various types of reference stations, the most common of which use filter-based gravimetric samplers to measure PM [34].

Low-cost sensors (LCSs) are used to collect real-time, high-resolution spatial and temporal air quality data. LCSs can measure PM, gas, or both. These sensors often consist of dense networks of multiple sensors, providing continuous monitoring that can identify areas of high air pollution within a town or city. Examples of LCSs are optical analyzers, which are popular PM sensors used in portable LCSs that use infrared or laser light to interact with particles and measure PM concentrations by number and size [34].

Satellite imagery can be utilized to analyze both PM and gaseous pollutants. By examining the amount of light that reaches the Earth's surface and the amount that is reflected off, satellites can measure the concentrations of different particles in the atmosphere. This method is called aerosol optical depth measuring and involves comparing these measurements with data on ozone concentrations and visibility [34].

In our research, we used an R-INLA implementation within R-studio. The spatial correlation between observations is computed using the SPDE technique, which discretizes the space by establishing a mesh that generates a fictitious set of neighbors across the research region. The inferences and predictions that we make will be significantly influenced by the mesh's design. In order to prevent outcomes from becoming sensitive to the mesh itself, it is crucial to build a decent mesh. Despite the fact that mesh construction varies

from case to case, there are certain recommendations for producing the best possible mesh. The size of the triangles in the inner zone must be appropriate, the outer bounds must be preserved, and the mesh must not create unexpected and unnatural patterns. The mesh, in certain cases, should also contain internal constraints, particularly when a barrier model is used. The barrier model is created by selecting appropriate polygons (triangles) and setting them as impassable barriers (e.g., walls). In the next step, the necessary elements for the model are defined, such as the covariance, the appropriate random effect, and the prediction function. At the end, according to the received model (supplied with data), we obtain a solution, which in, our case, due to transparency and data problems, we present as average prediction values in given areas. On this basis, we can draw conclusions about a given model for a given problem [29,35]. A diagram of the described algorithm is presented in Figure 4.

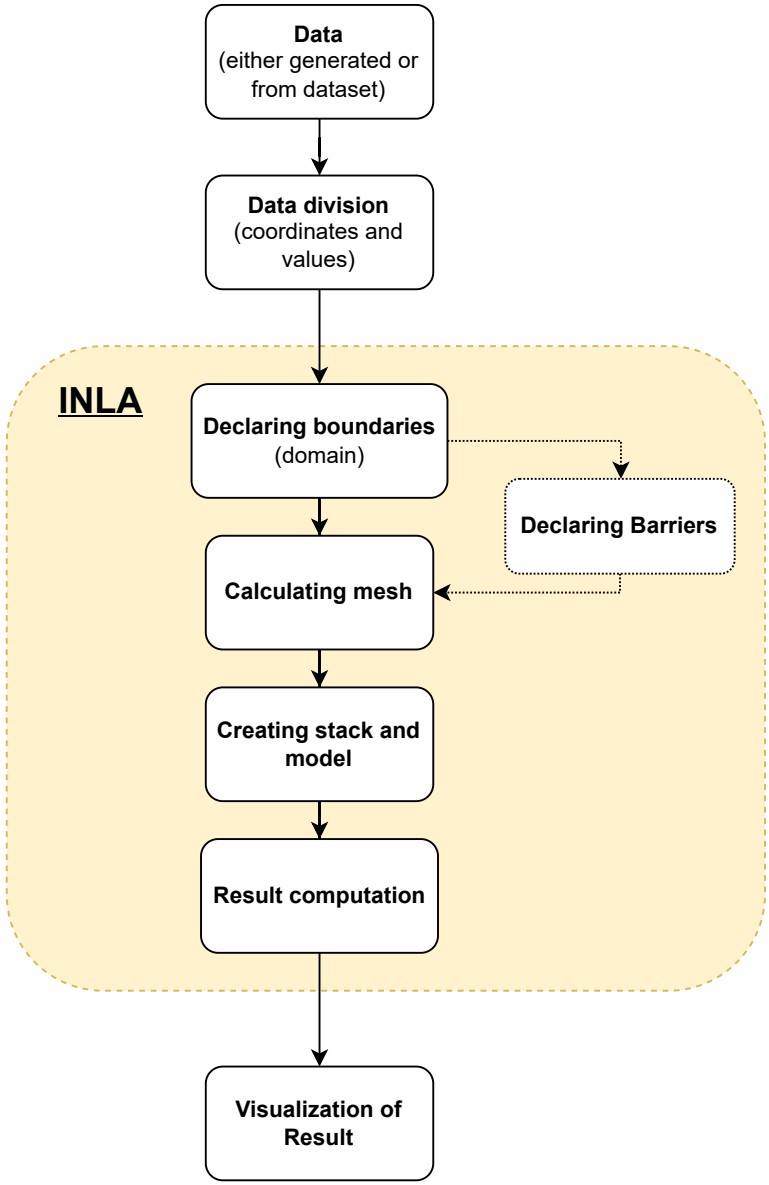

**Figure 4.** Diagrammatic representation of algorithm. First, we acquire data (which were generated or imported from a dataset) and then divide them into sets of coordinates and corresponding values. In INLA, we first declare the domain of our problem (boundaries), and if we are applying a barrier model, we also have to declare barriers. Next, we compute the mesh, which is used in the creation of the spatial model. Finally, the results are computed and visualized.

### 3. Results

The first two experiments were carried out in a three-room production hall defined for the model (Figure 1) on simulated data using the bivariate Gaussian distribution from Figure 2. In both cases, we determined the random positions of the PM10 measurement sensors in a defined space, where their values represent "measurements" from the data drawn using a bivariate Gaussian distribution. These sensor positions are shown in Figure 5. These 20 locations are inputs to the spatial model of air pollution in our defined factory. The locations of the sensors in the simulated space were based on a random selection of locations, with the condition that there must be sensors in each room.

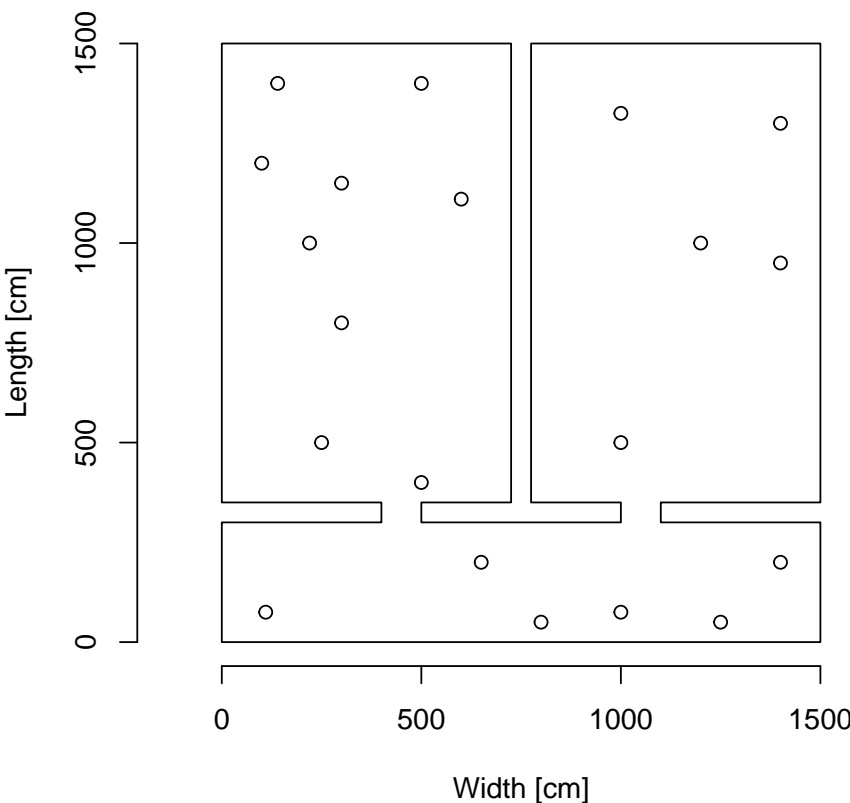

**Figure 5.** Selected positions of the sensors by randomized selection. Set of 20 points has values corresponding to the generated data using bivariate Gaussian distribution. Data from sensors are taken as inputs to both factory air pollution models.

The first step in creating a spatial model is to create the domain for which it will be calculated. In our case, it is, of course, a room with specific dimensions (15 m × 15 m). The mesh is the discretization of the domain (study area). The domain is divided up into small triangles. This is similar to creating a raster or a grid over space (reference). Such a mesh requires setting the appropriate values for the size of individual triangles constituting its interior (the place where all the data are located) as well as the appropriate buffer zone for the outer boundary. An equally important element is limiting the model only to the zone of considered data (in our case, the boundaries of the room). The mesh created in this way is shown in Figure 6, where the boundary separating the considered data area from the boundary is marked in blue, and the points with available data (simulated sensors) are in red. In the case of a general spatial model, we do not store information about internal wall barriers.

**Constrained refined Delaunay triangulation**

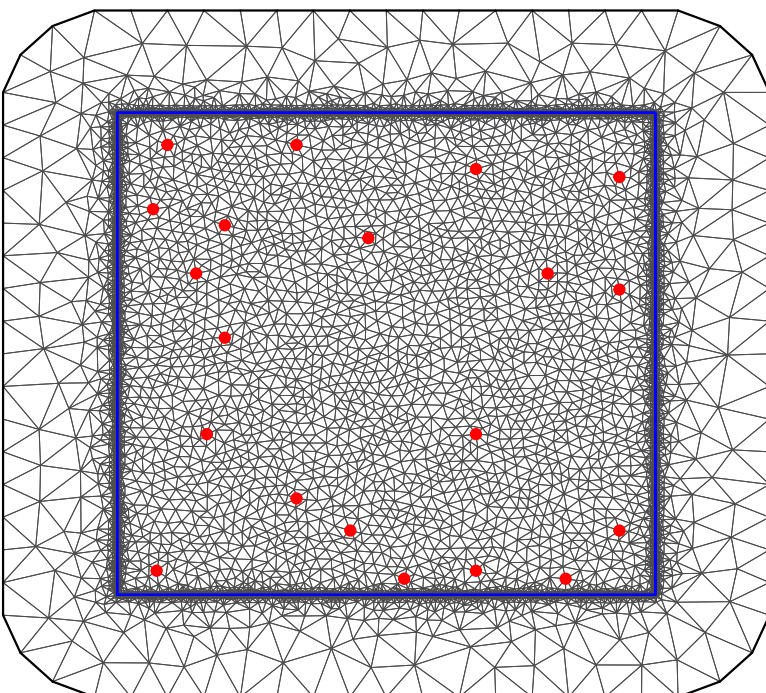

**Figure 6.** Mesh model for the first spatial modeling approach. The domain (inner section) is divided into small triangles, which allows them to connect to data from sensors. The outer boundary (buffer zone) is delimited by the size of the room (domain), marked with a blue line. In our case, we limit ourselves to the space of 15 m × 15 m only. Red dots represent positions of sensors.

The next stage of creating a spatial model in INLA is the creation of the so-called stack. Its first element is matrix A, mentioned earlier, whose main task is to connect information (data) at each point to its location in the previously created mesh. Moreover, it is a way of supplying covariates and random effects to INLA. For more complex spatial models, the stack is incredibly helpful, as the alternative is worse (we would have to construct the total model A matrix by hand). The last part is the definition of the spatial model component (random effect). In our case, the prior for the parameters/random effect is a Matérn prior with the hyperparameter range and marginal standard deviation. We start by defining the prior median for these hyperparameters. We also define the predictor for the model, which is a sum of model components and the observation likelihood.

The model created in this way allows the calculation of marginal distributions of hyperparameters, as well as marginal distributions of fixed effects. Plotting the posterior of the parameters in a random effect is much harder than plotting the hyperparameters because of the dimension of the parameters. So, as a result we do not plot the marginals, but instead we plot the summaries (like the mean). So, we present such a summary in the form of a spatial mean field (called a spatial estimate or spatial smoothing) as a solution to the problem. The graph in Figure 7 presents the average values obtained from the spatial model, with marked places representing simulated sensors. This spatial model is not a perfect solution to the problem of estimating the level of indoor air pollution, because it ignores obvious environmental obstacles, such as walls. This is important because isolated rooms at the bottom of the presented spatial plan of rooms should contain much lower average values in places separated by walls, and the only possible increases occur within the passages themselves.

**Result of spatial model**

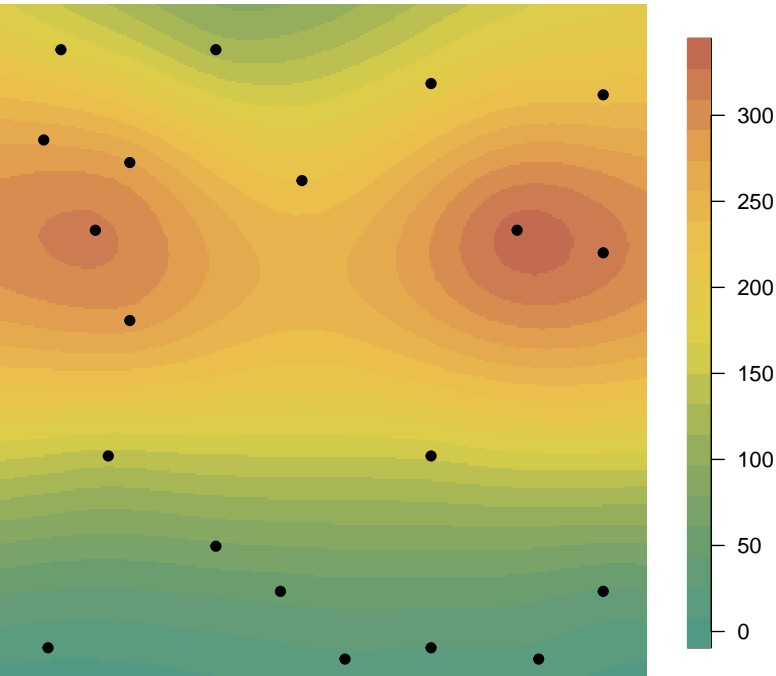

**Figure 7.** Result of spatial model without modeling space barriers. In the figure, the average pollution values for a given location in space are represented. Black dots represent positions of sensors. This model effectively reverses the randomly generated data from the bivariate normal distribution. It should be noted, however, that due to the lack of information about the walls of the building, it does not take into account, for example, the separation of the lower room, which makes it seem to have a much higher pollution factor than it would in practice.

In our second approach, we addressed the aforementioned imperfection of the spatial model for the production hall. The main difference is the implementation of an additional space barrier model, visible in Figure 8. It allows us to include information not only about the boundaries of the calculation domain but also about places that should be treated as impassable barriers. The created mesh model of space, of course, has the same limitations in regard to the domain of calculations indicated by the barrier model. The blue line in Figure 9 shows identical constraints to the barrier model. This approach allows us to include additional information for the model that should solve the problem of the previous approach.

The barrier model is subject to the same structure of settings as its previous version, i.e., the appropriate random effects, matrix A or predictor. Also, the result is presented in the same form as the average value of pollutants in space. However, in Figure 10, we see a difference in the form of the barriers involved. It is this factor that makes the results more reliable than the previous model. It is true that it does not perfectly represent the simulated data, but those data were only used to select a few values in space and not to calculate the quality of the model. In the case of the barrier approach, we clearly see that the model more reliably reflects the realistic spread of pollutants. The two rooms with sources of pollution have high values over the entire area. However, in the case of the isolated room, the inclusion of the walls allowed the model to change the prediction behavior and prevent air from spreading through the walls. This means that higher peaks of pollution values are only within the passages connecting individual rooms. Such results reflect reality better than the previous spatial model.

**The barrier region (in grey)**

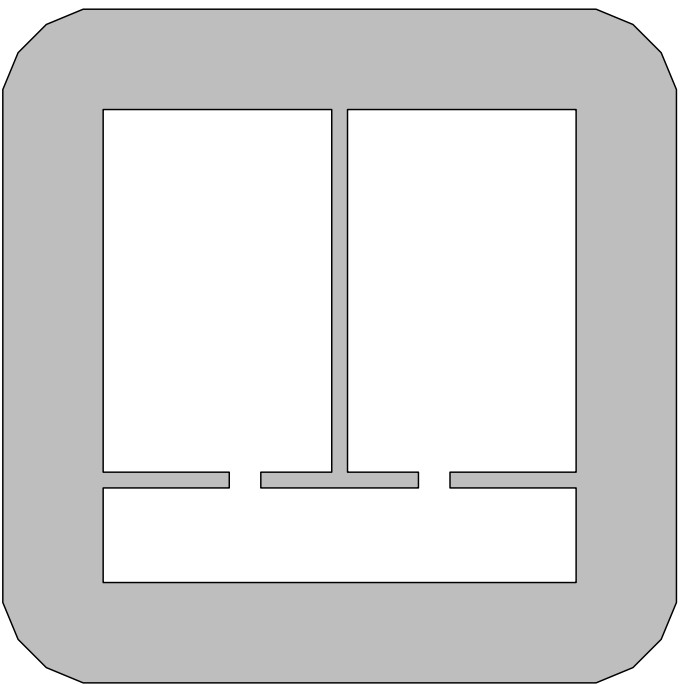

**Figure 8.** Barrier model for the considered production hall. It contains information about spaces inaccessible to the considered model (in our case, air pollution). It shows three rooms, two production rooms and one passage between them, as well as external and internal walls.

**Constrained refined Delaunay triangulation**

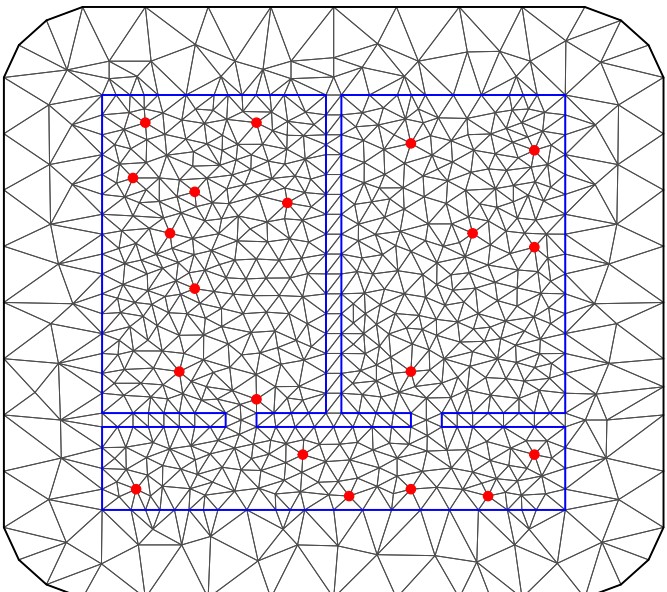

**Figure 9.** Mesh model for the second spatial modeling approach. The domain (inner section) is divided into small triangles, which allows them to connect to data from sensors. The inner zone (domain) is delimited by the location of barriers (walls) presented within barrier model from the outer boundary (buffer zone) and marked with a blue line. In our case, we limit ourselves to the space of 15 m × 15 m, only excluding walls. Red dots represent positions of sensors.

**Result of barrier model**

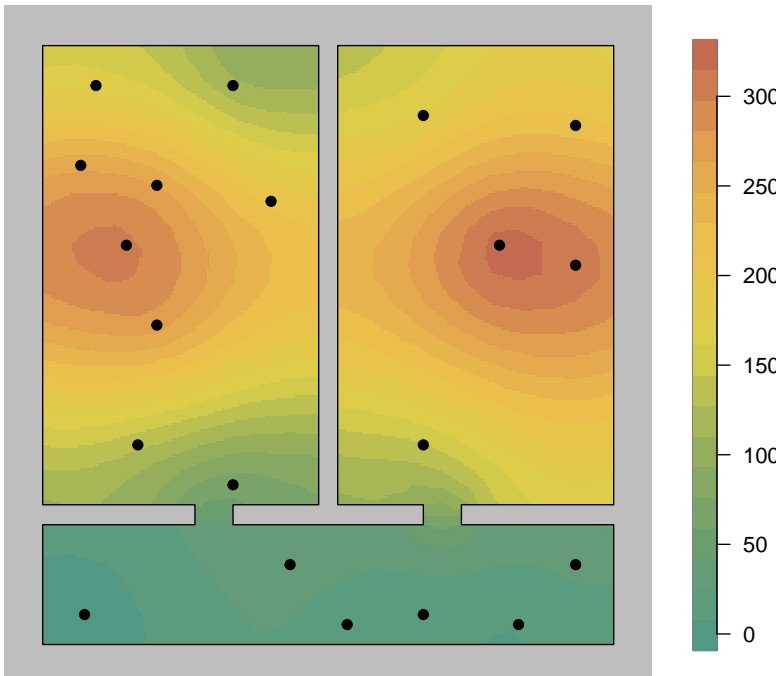

**Figure 10.** Result of spatial model with modeled space barriers. In the figure, the average pollution values for a given location in space are represented. Black dots represent positions of sensors. The result of this model is a good representation of the probable actual distribution of pollutants. Two production rooms, heavily polluted, do not differ significantly from those in the previous model. Separated rooms in the lower part have a better representation of average pollution values, taking into account higher values only in the area of passages and not in the whole space (where there are impassable walls).

The last model on which we conducted research was the spatial model of air pollution in the city of Krakow. As official professional sources have few research stations, we have merged the official data with the Airly database (which itself consists of many smaller databases and various sources and sensors). This allowed us to cover a sufficiently large area of Krakow (and its surroundings). In Figure 11, we have meshed our domain of consideration according to the previous procedure. Importantly, for better solutions at the edges of our final result, the domain area is larger than in the final considerations and has no clear separation from the buffer zone. As before, the red dots mark the locations of data from different sensors, from all databases together.

Based on information from the previously created spatial models representing air pollution, we set the priorities of our assumptions and predictors in a similar way. A noticeable difference is the lack of use of the barrier model (because it does not make sense for open airspaces) and not limiting the domain of considered data with the buffer zone. As mentioned earlier, the data are from midnight on 16 December 2022, and the average values calculated by the spatial model are included in Figure 12. In addition, characteristic places of the city (main streets, airport, etc.) have been marked in the background in order to improve orientation on the map. Judging from the results, the highest air pollution that day did not exceed 150 units, and its main accumulation was in the west and in the city center. Additionally, locations of stations that provided data are marked with dots, and the measuring stations of MIEP in particular are marked in blue.

**Constrained refined Delaunay triangulation**

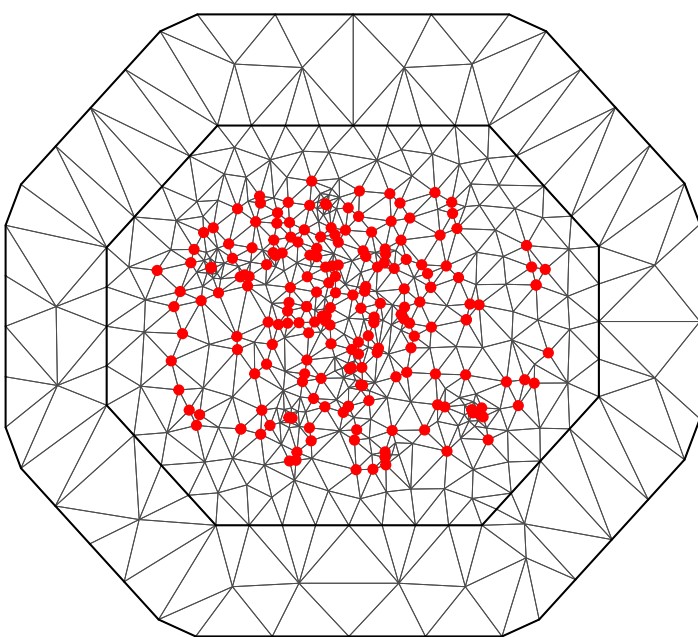

**Figure 11.** Mesh model for the spatial model of Krakow air pollution. The domain (inner section) is divided into small triangles, which allows them to connect to data from sensors. Outside, we can see the outer boundary (buffer zone). For this mesh, the domain of consideration is larger than the final result considered so as to be able to compute edge space information. Red dots represent positions of sensors from all datasets.

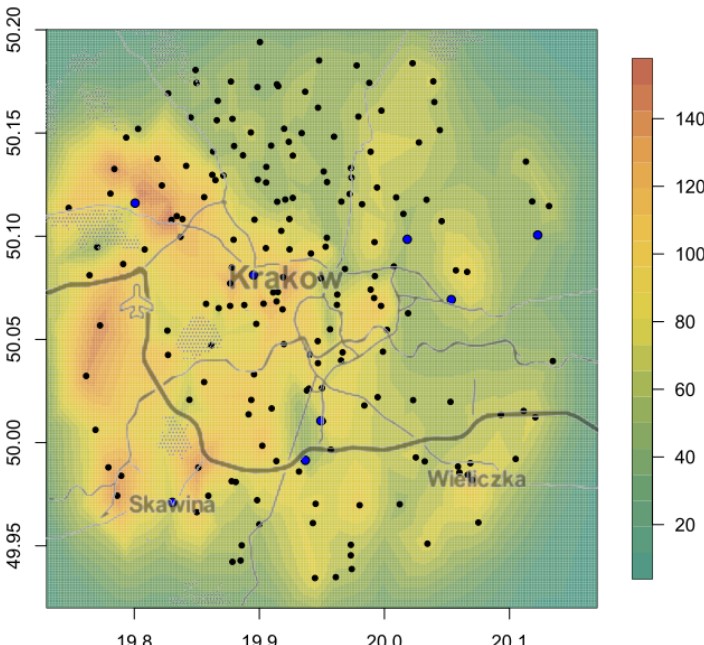

**Figure 12.** Result of spatial model of Krakow air pollution. In the figure, the average pollution values for a given location in space are represented. Black dots represent positions of sensors. Blue dots also represent positions of sensors, but only those from the MIEP dataset. In the background, there are elements that allow orientation in parts of the city (e.g., roads, airport). The maximum value of air pollution does not exceed 150, and its main concentration is in the western and central parts of the city. The *x*- and *y*-axes represent latitude and longitude values.

## 4. Discussion

The problem of air quality is a problem affecting many large and modern cities, as well as working spaces in factories. Its adverse effect on well-being and health is the key reason for solving it. However, before such measures are implemented, it is essential to understand the exact causes, sources, and behavior of polluted air in structures, whether in closed or open environments. Only then can appropriate countermeasures be taken. This is why it is so important to analyze the available data in advance and build appropriate mathematical models. The use of INLA in predicting the behavior of air pollution and detecting its sources is nothing new [5,13]. Some of the available studies even used a data fusion approach, as in our case. This is often necessary because even large cities, such as the examined Krakow, do not have a sufficiently dense infrastructure of official measuring stations. Another example of air pollution on a wide scale is presented in [36]. Research conducted during the COVID-19 pandemic examined the impact of carbon-based transport and industrial activity on air quality with the use of machine-learning techniques. The results provided by our algorithm, despite the structure of official stations not being sufficiently dense, may be relevant due to the use of data fusion from different collections and types of measurements. Using only the official measuring stations in the construction of the model could also be possible with the use of appropriate priors and INLA settings, but they would be burdened with high uncertainty. The use of data fusion allowed for the concentration of places where the information is reliable.

A different topic is pollution in workplaces, caused, for example, by the operation of machines. Detecting and preventing the causes is much simpler and less expensive compared to cities due to the scale of the problem. Separating the room with impurities is much easier. Detection, however, may prove to be a less trivial task due to, for example, a slight increase in contamination above the norm or contamination caused by a malfunction in the device. The problem of calculating air pollution exposure indoors can also be related to public spaces [37], taking into consideration human movement. It uses an activity-based travel demand model and low-cost air sensor network data, which leads to the use of different techniques than spatial modeling. A similar indoor problem can also be found in [38], where the research focuses on air quality in residential buildings. The topic is related to our problem, but it takes a different approach. It is based on a correlation with outdoor air pollution and takes into consideration the ventilation model, while ours provides results based on actual measurements, which can lead to better results, and the response within industrial structures. In the working environment of machines, our model can quickly recreate the current course of pollution in a two-dimensional space. It could be a very good addition to a system for warning about high pollution and detecting the place (source) of the failure, which may be difficult to locate using other sensors. Our model could discriminate between types of pollutants if other inputs were chosen. For this reason, it is worth analyzing information from sensors, especially with the use of barrier models, which can more accurately determine the cause and source of pollution, taking into account obstacles in the way.

## 5. Conclusions

The conclusion of this research in the field of spatial modeling is satisfactory. This is an early phase of research, the aim of which was to create a spatial model for indoor air quality and for open areas of the city. The first case was divided into two stages: a model without obstacles and a model with them (barrier model). Both models were based on the same source data, i.e., generated from a bivariate normal distribution. They were to simulate two rooms, each with a separate source of pollution (machine) and a connection (corridor).

In the first case, we can see that the model perfectly reflects the generated distribution, despite the small number of observation points (sensors). The maximum pollution values reach around 300, and the minimum values are in the vicinity, but greater than 0. Unfortunately, the aim of the model should be to create the best possible representation of indoor

air quality, taking into account, among other things, narrow passages, and not attempt to perfectly reproduce the pattern of generated data. For this reason, the second approach seems more appropriate. The barrier model allows us to take into account terrain obstacles, such as narrow spaces or walls. For this reason, it is much more suitable for modeling interior spaces. Thanks to these properties, it correctly detects, for example, a fenced-off lower corridor, in which the level of contamination is significantly reduced.

The case of the city of Krakow perfectly presents the operation of the model on a large scale of the city. On the day covered by the study, the level of pollution was concentrated around selected areas, which may be an appropriate place to consider a high source of pollution (e.g., an airport). It is equally important to use data fusion and to combine official information, including other measuring stations, not necessarily using identical measuring methods. This made it possible to precisely determine the level of pollution in space.

As a tool, INLA is perfect for solving these types of problems. The presented results clearly show the potential of this method, as well as areas for the further development of research. In the current, early stage, the algorithm does not have long-term history learning options, but adapting it to such a functionality may be a good step in improving our research. The same applies to extending the study of space from two to three dimensions. This would require the development of appropriate settings for the prediction model, prior, etc., but the INLA itself can accept data in three dimensions. So, the next step that we would like to take in the context of further research is to obtain more accurate models by, for example, taking into account heights for internal models and not using just two-dimensional coordinates. This will also allow us to consider information about spatial barriers that are not located at the entire height of room (e.g., furniture). In addition, with large-scale models such as cities, we could modify the model with additional information, such as a wind forecast, which will allow us to predict the level of pollution in the short and long term.

**Author Contributions:** Conceptualization, A.D. and J.B.; methodology, J.B.; resources, A.D. and J.B.; writing—original draft preparation, A.D.; writing—review and editing, A.D. and J.B.; visualization, A.D.; supervision, J.B.; project administration, J.B.; funding acquisition, J.B. All authors have read and agreed to the published version of the manuscript.

**Funding:** This research was funded by AGH's Research University Excellence Initiative under project "Interpretable methods of process diagnosis using statistics and machine learning" and by the Polish National Science Centre project "Process Fault Prediction and Detection", contract no. UMO-2021/41/B/ST7/03851. Part of work was funded by AGH's Research University Excellence Initiative under the project "Interpretable methods of process diagnosis using statistics and machine learning".

**Data Availability Statement:** Not applicable.

**Conflicts of Interest:** The authors declare no conflict of interest.

## Abbreviations

| | |
|---|---|
| INLA | Integrated nested Laplace approximation |
| MCMC | Markov chain Monte Carlo |
| GMRF | Gaussian Markov random field |
| MIEP | Main Inspectorate of Environmental Protection |
| PM | Particulate matter |
| LCS | Low-cost sensor |

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
