# Peer review of "Spatial Modeling of Air Pollution Using Data Fusion"

_electronics, doi:10.3390/electronics12153353_

Round 1

Reviewer 1 Report

The authors have made a spatial modelling of air pollution by data fusion. The article is very relevant and has a very rigorous methodology; it is also very well structured. I have some suggestions that would help to improve the quality of the article.

1. Justify why the results can be trusted even if the infrastructure of official measurement stations is not sufficiently dense. 

2. What advantages does the model presented by the authors have over other models for detecting pollution in workplaces caused by the operation of machines? 

3. Does the model distinguish between types of pollutants?

4. Under which criteria is the location of the sensors used based?

5. Can the programme be fed back on changes in ventilation levels or user habits?

6. The state of the art can be strengthened by taking into consideration impact articles such as :

- Lu, Y. (2021). Beyond air pollution at home: Assessment of personal exposure to PM2.5 using activity-based travel demand model and low-cost air sensor network data. Environmental Research, 201. https://doi.org/10.1016/j.envres.2021.111549

-Wijnands, J. S., Nice, K. A., Seneviratne, S., Thompson, J., & Stevenson, M. (2022). The impact of the COVID-19 pandemic on air pollution: A global assessment using machine learning techniques. Atmospheric Pollution Research, 13(6). https://doi.org/10.1016/j.apr.2022.101438

-Gonzalo, F. D. A., Griffin, M., Laskosky, J., Yost, P., & González-lezcano, R. A. (2022). Assessment of Indoor Air Quality in Residential Buildings of New England through Actual Data. Sustainability (Switzerland), 14(2). https://doi.org/10.3390/su14020739

7. It would be interesting to know if the model can infer the amount of TVOC in coordinates in space within an indoor area, so that the authors can show these values in the article; not only in horizontal planes; but also in vertical planes.

Author Response

  1. Justify why the results can be trusted even if the infrastructure of official measurement stations is not sufficiently dense. 

The results provided by our algorithm, despite the structure of official stations is not sufficiently dense, are relevant due to the use of data fusion from different collections and types of measurements. Only using of the official measuring stations in the construction of the model could also be possible with the use of appropriate priors and INLA settings, but they would be burdened with high uncertainty.

  1. What advantages does the model presented by the authors have over other models for detecting pollution in workplaces caused by the operation of machines? 

The use of data fusion allowed for the concentration of places where the information is reliable. In the working environment of machines, our model can quickly recreate the current course of pollution in two-dimensional space. It can be a very good addition to the system of warning about high pollution and detecting the place (source) of failures that may be difficult to locate using other sensors.

  1. Does the model distinguish between types of pollutants?

Our model could distinguish between types of pollutants if that kind of inputs were chosen. Currently all usage and setting was done with PM10 in mind.

  1. Under which criteria is the location of the sensors used based?

The location of the sensors in the simulated space was based on a random selection of locations, with the condition that there must be sensors in each room. The locations of the sensors in the city were based on actual coordinates.

  1. Can the programme be fed back on changes in ventilation levels or user habits?

In the current, early stage, the algorithm does not have long-term history learning options, but adapting it to such functionality may be a good step in improving our research.

  1. The state of the art can be strengthened by taking into consideration impact articles such as :

- Lu, Y. (2021). Beyond air pollution at home: Assessment of personal exposure to PM2.5 using activity-based travel demand model and low-cost air sensor network data. Environmental Research, 201. https://doi.org/10.1016/j.envres.2021.111549

-Wijnands, J. S., Nice, K. A., Seneviratne, S., Thompson, J., & Stevenson, M. (2022). The impact of the COVID-19 pandemic on air pollution: A global assessment using machine learning techniques. Atmospheric Pollution Research, 13(6). https://doi.org/10.1016/j.apr.2022.101438

-Gonzalo, F. D. A., Griffin, M., Laskosky, J., Yost, P., & González-lezcano, R. A. (2022). Assessment of Indoor Air Quality in Residential Buildings of New England through Actual Data. Sustainability (Switzerland), 14(2). https://doi.org/10.3390/su14020739

Thank you for the literature position you have sent. We used them to enrich our article.

  1. It would be interesting to know if the model can infer the amount of TVOC in coordinates in space within an indoor area, so that the authors can show these values in the article; not only in horizontal planes; but also in vertical planes.

Extending the study of space from 2 to 3 dimensions, would require the development of appropriate prediction model, prior etc. and proper settings of the R-INLA framework, but INLA itself can accept data in three dimensions. It is a great suggestion and we will try to apply it in the next iteration of algorithm.

Reviewer 2 Report

Reading the title of “Spatial modelling of air pollution using data fusion” one cannot help having high hopes for an extensive and broad scientific advance in air pollution modelling in all its layers of complexity. However, that is certainly not the case. The modelled case is too narrow and it does not capture the problem complexity.

The redaction is generally unsuitable for a research article, including subjective or unsubstantiated claims and even naivety.

The introduction is too short and it does not accomplish an introduction main goals with clarity, correctness and detail. It even has question marks in references.

A comprehensive related work investigation is missing.

The methodology is simple and straightforward, lacking a better explanation. Yet, more important is the absence of novelty in the research.

Results, its discussion, and conclusion are better that the previously mentioned but still far from being suggested for acceptance.

 -

Author Response

Reading the title of “Spatial modelling of air pollution using data fusion” one cannot help having high hopes for an extensive and broad scientific advance in air pollution modelling in all its layers of complexity. However, that is certainly not the case. The modelled case is too narrow and it does not capture the problem complexity.

We are grateful for your comments, and we agree that our approach does not capture the entirety of the phenomena, but that was never the intention. Our goal was to propose a method of simple data fusion in context of spatial statistical modelling.

The redaction is generally unsuitable for a research article, including subjective or unsubstantiated claims and even naivety.

We respectfully disagree with complaints about unsubstantiated claims, as in our opinion no such take place in the paper.

The introduction is too short and it does not accomplish an introduction main goals with clarity, correctness and detail. It even has question marks in references.

 Thank you for that suggestion, we have expanded the introduction, especially to clearly indicate our contribution.

A comprehensive related work investigation is missing.

  Thank you for that suggestion, we have significantly updated the discussion section.

The methodology is simple and straightforward, lacking a better explanation. Yet, more important is the absence of novelty in the research.

We have reorganized the methodology to make it more clear. We however respectfully disagree with complaint about lack of novelty, as in our paper we propose new approach of usiong certain statistical algorithms in technical context.

Results, its discussion, and conclusion are better that the previously mentioned but still far from being suggested for acceptance.

We respectfully disagree, as in our opinion it fits into the scope of the journal, especially that we have improved on discussion and conclusions.

Reviewer 3 Report

The manuscript entitled "Spatial modelling of air pollution using data fusion" needs some revisions in order to be ready for publication. In particular:

1. The abstract should be rewritten. At this point the most important seems to be INLA software. Also, some of the main results should be stated within the abstract.

2. P1L22: [ 2? ,3 ] remove questionmark.

3. The introduction at this point is very poor. It should be enriched with more information and bibliography.

4. Sections 2 and 3 is very solid and precise. But section 2 has to be part of the Introduction. Also section 3 should be part of Material and Methods due to the fact it is author's main methodological approach.

5. Figure 1 is too poor. Provide a new figure, and what is said in the caption move it within the figure.

6. Use passive voice within the whole manuscript.

7. Do not interrupt the text with images and tables (e.g. Figure 1, Figure 2 and Table 1.

8. Do not place 2 or even 3 images in the row without providing any text explaining to the reader after each image.

9. The Discussion at this point is very poor. Rewrite this section and enrich it with bibliography and compare your work to other studies. The current discussion of the scientific manuscript can be improved to enhance clarity and coherence. By refining the language and providing a clearer structure to the paragraph, the revised version conveys the intended message more effectively.

Minor editing of English language required

Author Response

We are grateful for your feedback, which allowed us to improve our article.

  1. In fact, one of the main goals is to explore and demonstrate the capabilities of the INLA framework, so the abstract has been revised to accommodate this and other suggestions.
  2. The question mark in the literature is a Latex interpretation error of third citation. We change the code, so it should not happen again.
  3. The introduction has been enriched with new bibliography, added more information about problem, clearly state our contribution.
  4. Thank you for the suggestion about chapters 2 and 3. They have been slightly modified and merged into the Introduction and Material and Methods accordingly.
  5. We tried to apply all corrections regarding the form of text and figures (especially the first figure).
  6. Thank you for the suggestion, but using the first person form applies more with our style concept.
  7. We applied some modification to the structure of images in order to present a clearer view for reader
  8. Same as 7
  9. The Discussion section has been changed. We added more examples of other, correlated work, as well as detailed explanation of research.

Reviewer 4 Report

Reviewer’s Report on the manuscript entitled:

Spatial modelling of air pollution using data fusion

The authors demonstrated INLA spatial modellings for air pollution, such as indoor air quality and for open areas of the city. Though the topic is important, the presentation and literature review must be  improved. In addition, I see many grammar and typo/style issues in the manuscript that should be checked and corrected. Please see below my comments.

Line 2. Grammar issue. “…One of such models is” not “are”. Please rephrase the whole sentence.

Line 3. Please define acronym INLA. All the acronyms must be defined the first time they appear.

Lines 5, 350, 353, etc. Please avoid using “you”. Either use “we” or third person in the entire manuscript.

Line 22. Please check the reference.

Line 30. “data” is plural. Please check and correct the verbs. Here, it should be “are” not “is”.

Line 44. The following review article show the spatial limitation coverage of various ground-base and satellite sensors and also platform that can be used for data fusion and processing, such as GEE which can be included here:

https://doi.org/10.1109/JSEN.2023.3246842

Line 50. Here please also include the following article for monitoring air quality using satellite data: https://doi.org/10.3390/atmos12121659

Line 51. Grammar issue. Please rephrase.

Line 85. Please highlight the main contributions of your work, preferably use bullet points.

Line 139. Grammar issue. It should be “…have been…”

Line 143. Grammar issue. It should be “First dataset comes from…”

Please add a flowchart in the method section to show the work flow of your research.

The Discussion section is short. Please mention the limitations and advantages of your research. Please also elaborate on your results in the light of other similar research.

Regards,

There are many grammar and typo/style issues in the manuscript that should be checked and corrected.

Author Response

Thank you for reviewing our article.

Text issues:
The points mentioned in the review have been fixed, and we have reviewed the entire article for minor errors. In addition, we edited the composition of the article to make it easier for the recipient to understand the topic. Abstract has been rewritten and introduction has been enriched with new bibliography positions and we state our contribution more clearly. The discussion section has been redrafted with new references and information about research area.

We highlighted the main contributions of your work with use of bullet points.

We added a flow chart as a graphical representation of our algorithm.

Also thank you for submitting important new literature positions that we could include in our article.

Round 2

Reviewer 1 Report

The authors have improved the article considerably. They have responded rigorously to all the comments made. I have nothing more to add.

Author Response

We express our gratitude for your response and the valuable suggestions provided, as they have enabled us to enhance our article significantly.

Reviewer 2 Report

I thank the Authors for the answer. The Authors disagree of most of my comments and, logically, did not perform significant changes according to those comments.

Moderate editing of English language required

Author Response

We sincerely appreciate your comments. While we may not have agreed with all of them, we have made substantial revisions of the article.

We acknowledged the need to enhance the introduction by providing more comprehensive background information and ensuring that all relevant references are included. In the revised manuscript, we expanded upon the existing introduction to provide a more thorough contextual framework for our research.

We rewritten the abstract to provide a clear and concise overview of our study, including the objectives, methods, key findings, and significance. We ensured that it accurately represents the content of the manuscript and engages readers' interest.

In the revised manuscript, we revied discussion and result section and focused on providing a comprehensive interpretation of the results, addressing limitations and potential problems, comparing our findings with previous studies, as well as proposing future research directions.

Additionally, we reviewed and incorporated any references that are pertinent to our study to ensure the inclusion of all relevant literature. We reviewed references in our revised manuscript to ensure their relevance to our research. Moreover, we enriched our article with many new references, which are relevant to the researched problem. Our aim was to strengthen the validity and reliability of our research by incorporating the most suitable references available.

We also understand the importance of effectively communicating our findings. We made the necessary revisions to enhance the clarity of our materials and methods section as well as result section. We provided more appropriate visual aids such as algorithm graph and corrected figures when necessary.

Although we may have had differing views on certain suggestions, we value and acknowledge your review.

Reviewer 3 Report

Dear authors,

thank you for addressing my comments.

Author Response

We appreciate your response and are grateful for your valuable suggestions, which have greatly contributed to the enhancement of our article.

Reviewer 4 Report

I thank the authors for addressing my comments. There are still many grammar/style/punctuation issues in the manuscript that should be carefully checked and corrected. I listed below some of them, but there could be more:

Lines 2-6 in the abstract. Unclear sentence. Please break it into two small sentences.

Line 27. The dot should be inserted after the references not before.

Line 38. Do you mean "Problems"?

Line 99. Please replace "In this paper we we want to investigate" with "In this paper, we investigate"

Lines 361 and 362. Unclear sentence. Please rephrase.

Regards,

There are many grammar/style/punctuation issues in the manuscript that should be carefully checked and corrected.

Author Response

Thank you for your comprehensive analysis and suggestions. We've implemented all of your recommendations, along with many more that we've been able to spot.